# Genome-Wide Identification and Expression Analysis of *FAR1/FHY3* Gene Family in Cucumber (*Cucumis sativus* L.)

**Xuelian Li, Yihua Li, Yali Qiao, Siting Lu, Kangding Yao, Chunlei Wang and Weibiao Liao ***

College of Horticulture, Gansu Agricultural University, Lanzhou 730070, China; 17609434640@163.com (X.L.);
liyihua_lyh@163.com (Y.L.); 18409487294@163.com (Y.Q.); 17339887987@163.com (S.L.);
19119882925@163.com (K.Y.); wangchunlei@gsau.edu.cn (C.W.)
* Correspondence: liaowb@gsau.edu.cn; Tel.: +86-138-9328-7942

**Abstract:** The *FAR1-RELATED SEQUENCE1* (*FAR1*) and *FAR-RED ELONGATED HYPOCOTYL3* (*FHY3*) gene family plays a crucial role in various physiological and developmental processes, including seed germination, photomorphogenesis, flowering and stress responses. However, genome analysis of *FAR1/FHY3* in cucumber (*Cucumis sativus* L.) has not been systemically investigated. In this study, 20 *FAR1/FHY3* genes in cucumber were identified. The 20 *FAR1/FHY3* members are randomly distributed on six chromosomes. The examination of subcellular localization indicated that the nucleus is the primary site where the 20 *FAR1/FHY3* members are predominantly found. The analysis of the phylogenetic tree further revealed that the *FAR1/FHY3* genes in cucumber are grouped into three distinct categories, exhibiting remarkable resemblance to the corresponding genes in other plant species. The analysis of cis-acting elements showed that most *FAR1/FHY3* genes contain a variety of hormones as well as stress-related and light response elements. Through scrutinizing the expression patterns in various tissues, it was discerned that these genes are prominently expressed in roots, stems and leaves, with roots exhibiting the highest level of expression. Additionally, the 20 cucumber *FAR1/FHY3* genes are all responsive to jasmonic acid methyl ester (Me-JA) and abscisic acid (ABA). *CsFAR6* and *CsFAR12* are significantly induced by Me-JA and ABA, respectively. *CsFAR13* positively responds to NaCl and PEG6000 stresses. *CsFAR11*, *CsFAR15* and *CsFAR13* are significantly induced by the dark. The findings presented in this study establish compelling support for the potential involvement of *FAR1/FHY3* genes in the growth, development and stress response of cucumbers. Moreover, these results serve as a solid basis for future investigations into the functional analysis of *FAR1/FHY3*.

**Keywords:** cucumber; photorespiration; function analysis; jasmonic acid methyl ester; abscisic acid; abiotic stress

## 1. Introduction

Plant growth and development are significantly influenced by light, which is a crucial environmental factor [1]. Higher plants have formed three types of photoreceptors in light morphology, such as phytochrome, cryptochrome and ultraviolet light (UV-B), which are used to sense different wavelengths of light [2,3]. Phytochromes directly or indirectly regulate multiple downstream transcription factors in response to red light and far-red light, thereby regulating downstream gene expression and affecting seed germination, de-luteinization, stem elongation, leaf expansion, plant shade avoidance and flowering induction [4]. FAR-RED IMPAIRED RESPONSE1 (FAR1) and FAR-RED ELONGATED HYPOCOTYL3 (FHY3), a group of homologous proteins derived from transposases, were originally identified in *Arabidopsis* as an important component of the phy A-mediated far-red light signaling pathway, a family of transcription factors found only in most an-giosperms [5]. The presence of *FAR1/FHY3* has been recognized in various eukaryotes, such as *Arabidopsis thaliana* L. [6], *Populus euphratica* Oliv. [7], *Eucalyptus robusta* Smith. [8],

*Arachis hypogaea* L. [9] and *Camellia sinensis* L. [10]. Current studies on the FAR1/FHY3 protein family mainly focus on Arabidopsis. In Arabidopsis, there are 14 members of the *FAR1/FHY3* family encoding 531 to 851 amino acids. In Arabidopsis, all members except *FAR-RELATED SEQUENCE 10* (*FRS10*) are expressed in leaves, stems, flowers and fruits [11]. Analysis of protein structure demonstrated that in Arabidopsis, FHY3 and FAR1 possess three primary functional domains. These include a $C_2H_2$ zinc finger domain situated at the N-terminal, enabling DNA-binding capabilities, a putative core transposase domain found in the middle-terminal regions and, finally, a SWIM zinc finger domain located at the C-terminal, which provides transcriptional activation activity within Arabidopsis [6,12]. Research on the evolution of the *FAR1/FHY3* family in Arabidopsis has indicated that it originated from transposases, which are plant-specific transcription factors [13,14].

The exploration of *FAR1/FHY3*'s significance in plant growth, development and resistance to both biotic and abiotic stress is progressively unfolding. *AtFHY3* and *AtFAR1* form homodimers or heterodimers that regulate various target genes particularly, which encode critical regulators of phy A translocating into the nucleus under far-red (FR) light [13]. In terms of light response, *AtFHY3* and *AtFAR1* play a crucial role in the response of mature plants to the dark-light transition [15]. FAR1 and FHY3, two genes that encode proteins associated with mutator-like transposases, directly stimulate the transcription of FAR-RED ELONGATED HYPOCOTYL1 (FHY1)/FHY1-LIKE (FHL) to regulate phy A signaling in response to far-red light circumstances by binding to the promoters of FHY1 and FHL [5]. In terms of growth and development, loss of FAR1 and FHY3 function led to increased accumulation of plant reactive oxygen species (ROS) and sensitivity to oxidative stress responses [16]. In addition, both FAR1 and FHY3 were positive regulators of chlorophyll biosynthesis [17]. In terms of stress response, during seed maturation and under unfavorable conditions, including drought and salinity, abscisic acid (ABA) accumulated to high levels and played important roles in seed dormancy, seedling growth and stomata movement [18]. They also bound to ABI5 promoters and activated their transcription, thereby mediating ABA signaling and abiotic stress responses [18].

Cucumber (*Cucumis sativus* L.) is a highly consumed vegetable globally due to its rich nutrients and ease of cooking. It is extensively used in various cuisines. Cucumber has numerous advantages, such as high yield, adaptability and easy cultivation, making it suitable for large-scale production and economically valuable in agriculture. Its industrial chain is also extensive, providing employment opportunities and economic benefits at various stages, including planting, harvesting, processing and marketing. Cucumbers are a globally cultivated vegetable crop. The growth and development of this vegetable are prone to unfavorable circumstances [19]. Until now, *FAR1/FHY3* genes in cucumber have not been well characterized and studied. Therefore, the current study seeks to first identify *FAR1/FHY3* gene members in cucumber. Then, the chromosomal location, physical and chemical properties, subcellular location, gene structure, protein structure and phylogenetic tree are analyzed. In addition, the expression pattern of *FAR1/FHY3* genes in different tissues and different treatments, including NaCl, dark, PEG6000, -ABA and methyl Me-JA, was investigated. Therefore, this study may provide a theoretical basis for further study of the biological function of *FAR1/FHY3* in plants.

## 2. Materials and Methods

### 2.1. Identification and Sequence Analysis of Cucumber FAR1/FHY3 Genes

First, the cucumber genome data and corresponding annotation files were downloaded from the EnsemblPlants database (ASM407v2) (https://plants.ensembl.org/index.html (accessed on 12 June 2023)) [20]. Here, 24 *FAR1/FHY3* gene members were already known in the ASM407v2 genome. Using the TBtools (Toolbox for Biologists) v2.003 software and the 'GXF Sequence extract' and 'Batch Translate CDS to Protein' functions in TBtools, FAR1/FHY3 protein sequence was extracted from database. Sequences of Arabidopsis FAR1/FHY3 genes were downloaded from the online database TAIR (https://www.

arabidopsis.org (accessed on 26 June 2023)). The structural domains of FAR1/FHY3 (PF03101, PF04434 and PF10551) were obtained from the PFAM database (http://pfam.xfam.org (accessed on 5 July 2023)). PFAM [21] and NCBI CDD (https://www.ncbi.nlm.nih.gov/Structure/bwrpsb/bwrpsb.cgi (accessed on 17 July 2023)) [22] databases were used to manually confirm that the candidate sequences had complete FAR1/FHY3 domains. Secondly, single BLAST of the collected sequences between cucumber and Arabidopsis was performed using the function of 'BLAST GUI Wrapper' in TBtools software, and the corresponding bidirectional BLAST [23] was performed using NCBI database (https://www.ncbi.nlm.nih.gov/ (accessed on 26 July 2023)). 'Blast Xml to Table 'function in TBtools software was then used to obtain possible gene family members of cucumber FAR1/FHY3. To determine further whether the identified protein belongs to the FAR1/FHY3 gene family, we used SMART (http://smart.embl-heidelberg.de (accessed on 23 July 2023)) to analyze the protein domain. Finally, we obtained the members of the cucumber FAR1/FHY3 gene family, and we used EXPASY (https://web.expasy.org/protparam/ (accessed on 26 July 2023)) to analyze the physical and chemical properties of these genes. The subcellular localization of protein was predicted using the online software PlantmPLoc (v2.0) (http://www.csbio.sjtu.edu.cn/bioinf/plant-multi/ (accessed on 28 July 2023)) [24] and NPSA (https://npsa-prabi.ibcp.fr/ (accessed on 29 July 2023)) online was used to predict the secondary structure of proteins.

### 2.2. Phylogenetic Analysis of Cucumber FAR1/FHY3 Genes

The FAR1/FHY3 protein sequences of cucumber, Arabidopsis and tomato were downloaded from the online database Ensembl [20]. The MEGA11 software (v2.010) was used to construct a phylogenetic tree of 61 FAR1/FHY3 protein sequences by applying the Maximum Likelihood method [25]. The bootstrap replication value was set as 1000, and the other parameters remained constant. The Evolview (https://www.evolgenius.info/evolview-v2 (accessed on 21 June 2023)) website was used to improve the appearance of the evolutionary tree.

### 2.3. Gene Structure and Chromosomal Localization Analysis of Cucumber FAR1/FHY3 Genes

The gene structure of each member of *FAR1/FHY3* was analyzed using the 'Visualize Gene Structure (from GTF/GFF3 File)' function in TBtools software. The cucumber GFF3 file was downloaded from the online database. The chromosomal position distribution of cucumber *FAR1/FHY3* genes was analyzed using the "Gene Location Visualize from GTF/GFF" function of TBtools software [26]. The *FAR1/FHY3* gene members were renamed according to their chromosome distributions.

### 2.4. Gene Structure and Conserved Motif Analysis of Cucumber FAR1/FHY3 Genes

The distribution of the conserved motifs based on amino acid sequence was conducted with the online MEME (http://meme-suite.org/tools/meme (accessed on 23 June 2023)) website [27]. The maximum number of Motif discoveries was set to 10, whereas other parameters were the default values. Then, the corresponding motif information and the evolutionary tree information of cucumber FAR1/FHY3 family derived from MEGA11 were combined to be analyzed via the 'gene structure view (Advances)' function of TBtools software to visualize the conserved motifs of cucumber *FAR1/FHY3* members.

### 2.5. Cis-Acting Element Analysis of Cucumber FAR1/FHY3 Genes

The 2000 bp DNA sequences in the cucumber *FAR1/FHY3* gene upstream region were screened as promoter sequences using Tbtools software. The PlantCARE database [28] was used to search for cis-regulatory elements in cucumber gene promoter region to study the role of genes in hormonal response and abiotic stress.

### 2.6. Collinearity Analysis of Cucumber FAR1/FHY3 Genes

The online database ensemble was used to determine the genetic relationship between Arabidopsis and cucumber. The genome files (FASTA) and genomic annotation file (GFF3) of the related cucumber *FAR1/FHY3* gene families were downloaded, and the collinearity analysis was performed by using the "Text Merge for MCScanX" function in TBtools software. Finally, the results were visualized by 'Multiple Systeny Plot' function in TBtools software [26].

### 2.7. Plant Materials, Cultivation Conditions and Treatments

Cucumber (*C. sativus* L. 'Xin Chun 4') seeds were soaked in water at 55 °C for about 4 h, then were placed on damp filter paper where they germinated overnight in the incubator at 25 °C. Seedlings were transferred into hydroponic boxes once the cotyledon fully unfolded. The boxes containing Yamazaki cucumber nutrient solution were placed in plant incubators at 25 °C, a light intensity of 200 $\mu$mol m$^{-2}$ s$^{-1}$ and a photoperiod of 14 h light/10 h dark [29]. Yamazaki cucumber nutrient solution [30] was used and replaced every 3 d. For drought, salt, ABA and Me-JA treatments, seedlings were grown in a 1/2 nutrient solution containing 8% (*w/v*) PEG6000, 50 mM NaCl, 100 $\mu$M ABA, 100 $\mu$M and Me-JA, respectively. These concentrations were selected based on our previous study [31]. For dark treatment, the seedlings were moved to a dark growth chamber and placed in 1/2 nutrient solution. Leaf samples were collected for qRT-PCR analysis after treatment for 6, 12, 24 and 48 h and were stored at $-80$ °C [32]. For the analysis of spatial expression patterns of cucumber *FAR1/FHY3* genes, the roots, stems and leaves of untreated seedlings were collected at the two-leaf stage and were stored at $-80$ °C. There were three biological replications with 8 cucumber seedlings in each replication collected under the identical experimental condition.

### 2.8. RNA Isolation and qRT-PCR Analysis

Total RNA was extracted from different tissues and treated with MiniBEST plant RNA Extraction Kit (Takara, Dalian, China). The FastQuant (Tiangen, Beijing, China) Synthesis Kit was used to synthesize First Strand cDNA fragments according to the manufacturer's method. qRT-PCR was performed using the SuperReal PreMix Plus kit (TIANGEN, Beijing, China) and Roche LightCycler instruments (Roche Applied Science, Penzberg, Germany). The CDS sequences of CsFAR genes were input into the homepage of Shanghai Biology Company (Shanghai, China) for online primer design [33] (Table 1), and then the primer sequences were synthesized. CsActin was used as an internal reference gene [34]. The amplification system contained 2 $\mu$L cDNA, 0.6 $\mu$L 10 $\mu$M upstream primers, 0.6 $\mu$L 10 $\mu$M downstream primers, 10 $\mu$L 2 $\times$ SuperReal PreMix Plus and 6.8 $\mu$L RNase-free ddH$_2$O. qRT-PCR cycle conditions included: 95 °C for 15 min, 90 °C for 10 s and 60 °C for 20 s, 40 cycles. There were three biological replicates per treatment. The cucumber FAR1/FHY3 gene was used to normalize relative expression levels. The $2^{-\Delta\Delta Ct}$ method [35] was used to calculate the relative expression.

**Table 1.** The sequences of primers used for qRT-PCR.

| Gene Name | Forward Primer Sequence (5$'$-3$'$) | Reverse Primer Sequence (5$'$-3$'$) |
|---|---|---|
| *CsFAR1* | TGGCAAACGCTTGTTGACAG | TGCAGGTATCCATCGACTGC |
| *CsFAR2* | CTGTTGTGGCCTCGTCAAGA | AACAGCCATGTGGTCAGGAG |
| *CsFAR3* | CCTCAAGAACTCAAGACCGTGATG | TCCTTCCACCCTCTCGCATAATC |
| *CsFAR4* | TGTCTGCTCCAACTGCCAAGG | TTTCAACATCTTCCCACACCAACTG |
| *CsFAR5* | TCCGAACGACGACGCCTTC | CCATCTTCCTCAACCAATTCTCCAC |
| *CsFAR6* | TGGATGTTGATGAAGGAGAGTTTGG | TTTGAATCTGTGAGTTGTGCTTGAC |
| *CsFAR7* | GCTTGCGTGACGATGATAAGG | GAACGGTGAATCCAACACGC |
| *CsFAR8* | TGTGAGGCAGAGGGCAGAAAC | CCAGGTGAGAGAGGATGCGTATG |

**Table 1.** *Cont.*

| Gene Name | Forward Primer Sequence (5′-3′) | Reverse Primer Sequence (5′-3′) |
|---|---|---|
| *CsFAR9* | CCAGGTGAGAGAGGATGCGTATG | CCTTGAATGCTCTGTTGGCTGAC |
| *CsFAR10* | ATCGGCAGCTCATGGTCTACTC | CTGGCTCGCAATTCACCTTCC |
| *CsFAR11* | CTTGGGTTCTGTTGGTTTTGTTGAC | CCGAATCCACCTCCACCTCTG |
| *CsFAR12* | CTTCGGGAGGAGAATTTGAGATCG | GTAGTAGGCAGTTCCATCGCTATTG |
| *CsFAR13* | TGCCGTGCTGCCTCCATC | GCCGTTCCACCATTCGTTGAG |
| *CsFAR14* | AGAGATAGGAAATCAGTGCGTTGTG | GGAAGAAGGCGAACTTGGTCATC |
| *CsFAR15* | TTGAGAAGAGGTGGCAGAAGTTG | CCATTCGTGAAGATGTGCATAAGC |
| *CsFAR16* | CCTCTCAGCAGTCTTGGTGG | TGGCAGCTTCTTCAGACTCG |
| *CsFAR17* | GTGAAGAGTGAGACAGTGCCATC | GTGAAGAGTGAGACAGTGCCATC |
| *CsFAR18* | GGAGTTTGAATCTGAGGAGTCTGTC | CCGCACCATCACGCATCG |
| *CsFAR19* | GCCATTATTACCGACGACATTGC | CGTGTCATCAAGGTCAGGAAGAG |
| *CsFAR20* | ACCCTCTCTTACCAGTGAGCAATC | CTTCCTTCCATGATGACCACCTAAC |
| *CsActin* | TGGACTCTGGTGATGGTGTTA | CAATGAGGGATGGCTGGAAAA |

## 3. Results

### 3.1. Identification and Characterization of FAR1/FHY3 Genes in Cucumber

In this study, 20 cucumber *FAR1/FHY3* genes were finally identified, named *CsFAR1* to *CsFAR20* according to the location of the gene on the chromosome (Table 2). As shown in Figure 1, the *CSFAR* genes are distributed unevenly on six chromosomes, and the number of genes on each chromosome is independent of chromosome size. The amino acid lengths range from 222 to 876 aa. *CsFAR18* has the smallest amino acid length and molecular weight; *CsFAR2* has the largest amino acid length and molecular weight. Furthermore, the isoelectric point changes from 5.08 (*CsFAR5*) to 8.94 (*CsFAR18*). The instability index changes from 33.10 (*CsFAR19*) to 59.49 (*CsFAR4*), with the instability index of *CsFAR17*, *CsFAR19* and *CsFAR20* less than 40, indicating that they are stable proteins, and the rest are unstable proteins. The aliphatic index changes from 68.18 (*CsFAR10*) to 91.00 (*CsFAR17*). In addition, CsFAR3, CsFAR14, CsFAR16 and CsFAR18 proteins were predictably located in the chloroplast and nucleus. *CsFAR6*, *CsFAR8*, *CsFAR10*, *CsFAR12* and *CsFAR17* were predictably located in the chloroplast. *CsFAR4*, *CsFAR11* and *CsFAR19* were predictably located in the chloroplasts and cytoplasm, nuclei and vacuoles, cell membranes and cell walls, and the rest were in the nucleus (Table 2). Upon examining the secondary structure, it was evident that cucumber proteins FAR1/FHY3 consist of an a-helix, a random coil, an extended strand and a β-turn (Table 3).

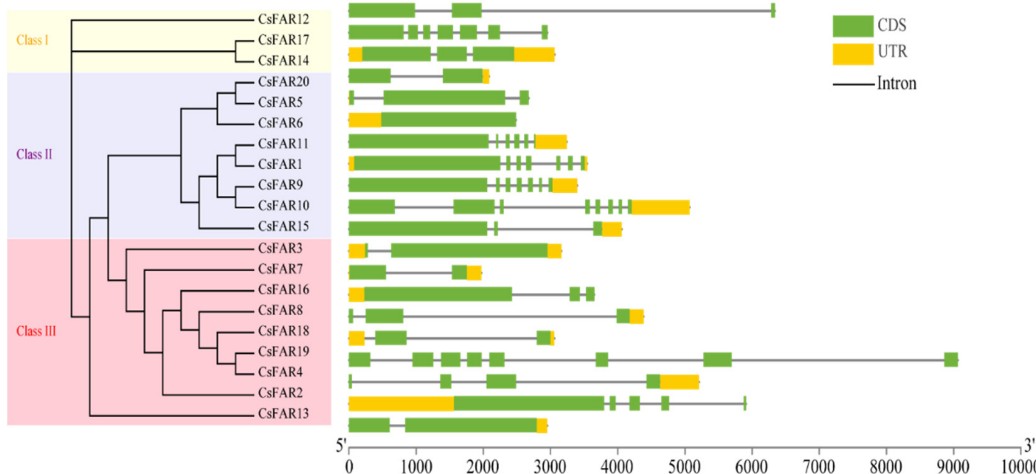

**Figure 1.** Structure analysis of the *FAR1/FHY3* gene in cucumber. MEGA11 was employed to construct the evolutionary tree utilizing the complete cucumber FAR1/FHY3 protein sequences. Furthermore, TBtools software was utilized to generate the exon-intron diagram for cucumber *FAR1/FHY3* genes.

Table 2. Characterization of *CsFAR1/FHT3* transcription factors in cucumber.

| Gene | Gene ID | Chr. No. | Chr. Location | Length (aa) | Mol. Wt. (kDa) | pI | Instability Index | Aliphatic Index | Grand Average of Hydropathicity | Subcellular Localization |
|---|---|---|---|---|---|---|---|---|---|---|
| *CsFAR1* | Csa_1G057030 | 1 | 6289393–6292939 | 855 | 98,227.11 | 8.53 | 55.46 | 69.30 | −0.577 | Nucleus. |
| *CsFAR2* | Csa_1G600950 | 1 | 23317773–23323685 | 876 | 99,141.71 | 5.94 | 48.19 | 73.38 | −0.443 | Nucleus. |
| *CsFAR3* | Csa_3G150180 | 3 | 10205529–10208691 | 788 | 90,617.81 | 8.31 | 47.68 | 74.51 | −0.464 | Chloroplast. Nucleus. |
| *CsFAR4* | Csa_3G811580 | 3 | 31128753–31133962 | 283 | 32,739.57 | 7.65 | 59.49 | 71.98 | −0.575 | Chloroplast. Cytoplasm. |
| *CsFAR5* | Csa_4G006400 | 4 | 1108119–1110798 | 672 | 77,927.58 | 5.08 | 42.49 | 75.42 | −0.523 | Nucleus. |
| *CsFAR6* | Csa_4G006410 | 4 | 1113075–1115564 | 669 | 77,316.64 | 5.61 | 40.80 | 82.57 | −0.407 | Chloroplast. |
| *CsFAR7* | Csa_4G015740 | 4 | 2030697–2032669 | 255 | 29,762.52 | 5.82 | 54.49 | 70.27 | −0.863 | Nucleus. |
| *CsFAR8* | Csa_4G017090 | 4 | 2263706–2268092 | 270 | 31,065.94 | 6.33 | 46.67 | 82.63 | −0.688 | Chloroplast. |
| *CsFAR9* | Csa_4G056770 | 4 | 4882886–4886283 | 808 | 94,208 | 8.17 | 54.71 | 73.02 | −0.588 | Nucleus. |
| *CsFAR10* | Csa_4G056780 | 4 | 4888663–4893732 | 555 | 62,862.94 | 6.99 | 51.83 | 68.18 | −0.661 | Chloroplast. |
| *CsFAR11* | Csa_4G618450 | 4 | 19741128–19744369 | 775 | 89,318.44 | 7.61 | 45.75 | 70.34 | −0.505 | Nucleus. Vacuole. |
| *CsFAR12* | Csa_5G003630 | 5 | 291625–297967 | 494 | 56,958.92 | 6.9 | 49.22 | 76.64 | −0.428 | Chloroplast. |
| *CsFAR13* | Csa_5G175740 | 5 | 7336329–7339280 | 855 | 98,441.65 | 8.23 | 40.30 | 77.44 | −0.364 | Nucleus. |
| *CsFAR14* | Csa_6G511070 | 6 | 26338871–26341932 | 692 | 80,019.07 | 6.01 | 44.80 | 81.16 | −0.326 | Chloroplast. Nucleus. |
| *CsFAR15* | Csa_6G538640 | 6 | 28958065–28962125 | 747 | 86,692.18 | 6.33 | 48.60 | 72.58 | −0.536 | Nucleus. |
| *CsFAR16* | Csa_7G029420 | 7 | 1561910–1565564 | 826 | 95,151.41 | 6.32 | 52.01 | 73.85 | −0.523 | Chloroplast. Nucleus. |
| *CsFAR17* | Csa_7G031740 | 7 | 1799957–1802915 | 602 | 68,955.26 | 6.64 | 39.13 | 91.00 | −0.251 | Chloroplast. |
| *CsFAR18* | Csa_7G372900 | 7 | 13359283–13362340 | 222 | 25,360.79 | 8.94 | 50.36 | 72.88 | −0.632 | Chloroplast. Nucleus. |
| *CsFAR19* | Csa_7G375750 | 7 | 13662027–13671093 | 725 | 80,007.84 | 6.22 | 33.10 | 86.15 | −0.255 | Cell membrane. Cell wall. |
| *CsFAR20* | Csa_7G432410 | 7 | 17215746–17217833 | 404 | 46,899.12 | 8.60 | 39.92 | 70.64 | −0.656 | Nucleus. |

Note: pI, isoelectric point. Mol. Wt., molecular weight.

**Table 3.** The secondary structures of CsFAR1/FHY3 proteins.

| Protein | Alpha Helix (%) | Beta Turn (%) | Random Coil (%) | Extended Strand (%) |
| --- | --- | --- | --- | --- |
| CsFAR1 | 40.47 | 2.92 | 45.61 | 10.99 |
| CsFAR2 | 40.41 | 4.11 | 40.87 | 14.61 |
| CsFAR3 | 44.8 | 4.44 | 37.94 | 12.82 |
| CsFAR4 | 31.80 | 6.01 | 41.70 | 20.49 |
| CsFAR5 | 42.41 | 4.61 | 37.65 | 15.33 |
| CsFAR6 | 45.29 | 4.48 | 35.87 | 14.35 |
| CsFAR7 | 40.39 | 5.10 | 38.43 | 16.08 |
| CsFAR8 | 40.37 | 4.07 | 40.74 | 14.81 |
| CsFAR9 | 43.44 | 2.97 | 41.71 | 11.88 |
| CsFAR10 | 38.02 | 3.42 | 47.75 | 10.81 |
| CsFAR11 | 40.52 | 2.97 | 43.23 | 13.29 |
| CsFAR12 | 42.91 | 6.48 | 35.22 | 15.38 |
| CsFAR13 | 46.67 | 4.91 | 33.45 | 14.97 |
| CsFAR14 | 46.24 | 4.05 | 35.12 | 14.6 |
| CsFAR15 | 42.44 | 2.81 | 42.30 | 12.45 |
| CsFAR16 | 41.77 | 4.36 | 39.95 | 13.92 |
| CsFAR17 | 46.35 | 4.98 | 36.05 | 12.62 |
| CsFAR18 | 39.19 | 5.86 | 38.74 | 16.22 |
| CsFAR19 | 23.17 | 5.52 | 41.93 | 29.38 |
| CsFAR20 | 38.37 | 5.20 | 42.33 | 14.11 |

*3.2. Genomic Structure and Protein Domain Analysis of FAR1/FHY3 Members in Cucumber*

Phylogenetic analysis data revealed the classification of cucumber FAR1/FHY3 proteins into three distinct categories (Figure 1). In detail, CsFAR12, CsFAR17 and CsFAR14 were classified into Class I; CsFAR20, CsFAR5, CsFAR6, CsFAR11, CsFAR1, CsFAR9, CsFAR10 and CsFAR15 were classified into Class II; and CsFAR3, CsFAR7, CsFAR16, CsFAR8, CsFAR18, CsFAR19, CsFAR4, CsFAR2 and CsFAR13 were classified into Class III. *CsFAR12*, *CsFAR14* and *CsFAR17* of Class I possess three, seven and three introns, respectively. *CsFAR20*, *CsFAR5*, *CsFAR6*, *CsFAR11*, *CsFAR1*, *CsFAR9*, *CsFAR10* and *CsFAR15* of Class II possess two, three, one, six, seven, seven, eight and three introns, respectively. In Class III, *CsFAR3*, *CsFAR7*, *CsFAR18* and *CsFAR13* have two introns; *CsFAR16* and *CsFAR8* have three introns; *CsFAR19*, *CsFAR4* and *CsFAR2* contain eight, four and five introns, respectively (Figure 1).

*3.3. Conserved Motifs of Cucumber FAR1/FHY3 Proteins*

Conserved motif analysis uncovered a total of 10 identical motifs in the cucumber's FAR1/FHY3 proteins (Figure 2). Table 4 presents a list of conserved motifs, including their sequence information. The length of each motif varies from 18 to 50 amino acids. Using TBtools software, we found that the cucumber FAR1/FHY3 proteins belonging to the same subfamily in the evolutionary tree contain a similar or identical composition of motifs. As a result, 10 conserved motifs were found, namely motifs 1–10. Most of the FAR1/FHY3 proteins harbor 10 common motifs with the order of 1, 10, 4, 6, 2, 8, 5, 7, 3 and 9, while some FAR1/FHY3 proteins only harbor part of those motifs with specific motifs missing. For example, CsFAR7, CsFAR8, CsFAR18 and CsFAR4 only contain motifs 1, 10 and 4. In Class I, CsFAR12 lost motif 7, motif 3 and motif 9; CsFAR17 lost motif 6, motif 8, motif 3 and motif 9; and CsFAR14 lost specific motif 9. In Class II, CsFAR20 and CsFAR10 lost motif 6, motif 2, motif 8 and motif 5, whereas CsFAR5 lost motif 3. In Class III, CsFAR3 had three repeating motifs (motif 1, motif 10, motif 4), whereas CsFAR19 lost motif 2, motif 8, motif 5, motif 7, motif 3 and motif 9. Generally speaking, for the motif feature of each class in cucumber, Class III is more complex in structure, followed by Class II and Class I (Figure 2).

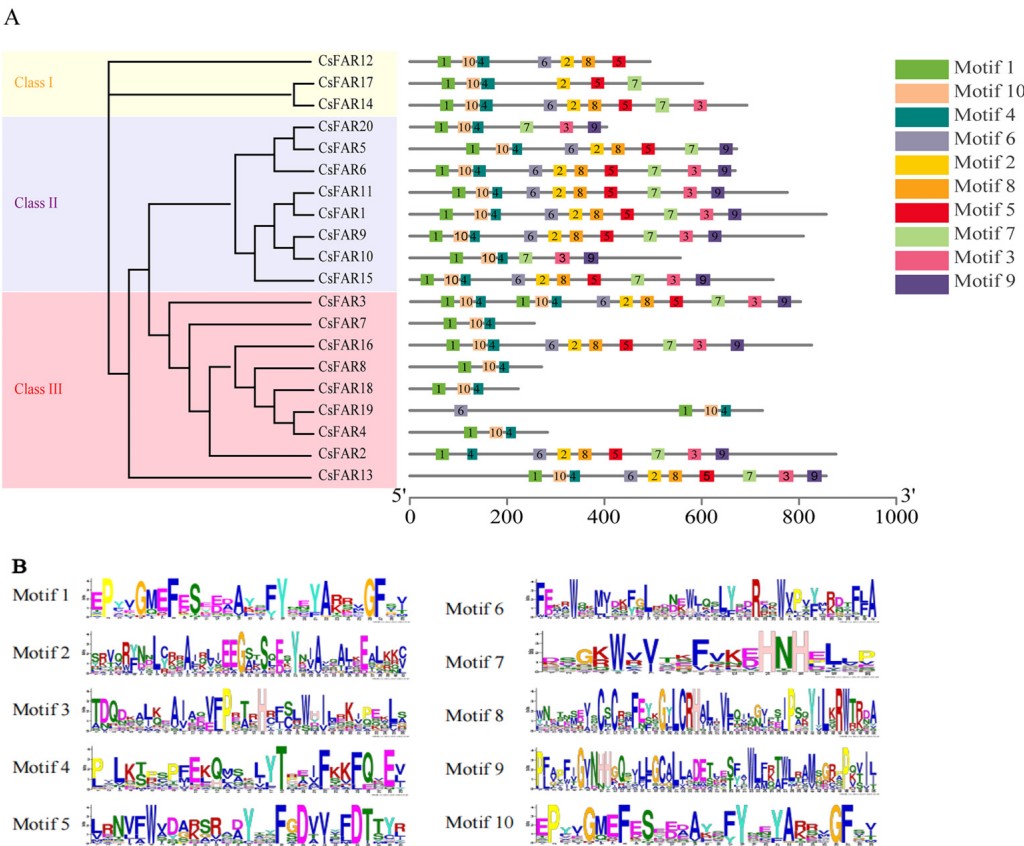

**Figure 2.** The distribution and composition of FAR1/FHY3 proteins in cucumber pertaining to motif occurrence. (**A**) Distinct motifs are represented by colored boxes. (**B**) Sequential stacks of letters illustrate the amino acid sequences associated with each motif. The overall height of the stack signifies the information content in bits of the respective amino acid at each position of the motif. The specific height of each letter within the stack is calculated by multiplying the probability of occurrence at that position and the total information content of the stack. Width and bits are depicted on the X and Y axes, respectively.

**Table 4.** Details of the 10 conserved motifs of cucumber FAR1/FHY3 proteins.

| Motif | Width (aa) | Motif Sequence |
|---|---|---|
| Motif 1 | 29 | EPYVGMEFESEEDAYEFYNEYARRVGFSV |
| Motif 2 | 50 | PFAPFIGVNHHGQSVLLGCALLADETLESFAWLFKTWLRAMSGRPPKTIJ |
| Motif 3 | 50 | WNKSNSEVSCSCRLFEYKGYLCRHALIVLQILGIKSJPSQYILKRWTRBA |
| Motif 4 | 20 | DSGKWVVTKFVKEHNHELLP |
| Motif 5 | 41 | FEKRWQKMVDKFGLRDBEWJQSLYSDREKWVPVYLRDTFLA |
| Motif 6 | 29 | LRNVFWVDAKSRADYSYFGDVVYFDTTYR |
| Motif 7 | 29 | PVLKSPSPFEKQMAKLYTHEIFKKFQVEV |
| Motif 8 | 36 | TDQDKALKEAIAEVFPETRHRFSLWHILEKIPEKLS |
| Motif 9 | 41 | SRVQRYNNLCRRAIKLIEEGSLSQESYNIALZALEEALKKC |
| Motif 10 | 18 | RPRPSTRTGCKAMMHVKK |

*3.4. Chromosomal Location of Cucumber FAR1/FHY3 Genes*

According to the results of chromosome localization, 20 *FAR1/FHY3* genes are randomly located on six cucumber chromosomes (Figure 3). Chromosomes 5, 6 and 7 harbor Class I *FAR1/FHY3* genes, whereas chromosomes 1, 4, 6 and 7 carry Class II *FAR1/FHY3* genes. More than half of the *FAR1/FHY3* genes in Class II are located on chromosome 4, whereas the genes for Class III are mainly distributed on chromosomes 1, 3, 4, 5 and 7. Hence, the dispersion of *FAR1/FHY3* genes across chromosomes appears to be fairly

uniform. Nonetheless, distinct groups display a tendency to localize on particular chromosomes (Figure 3).

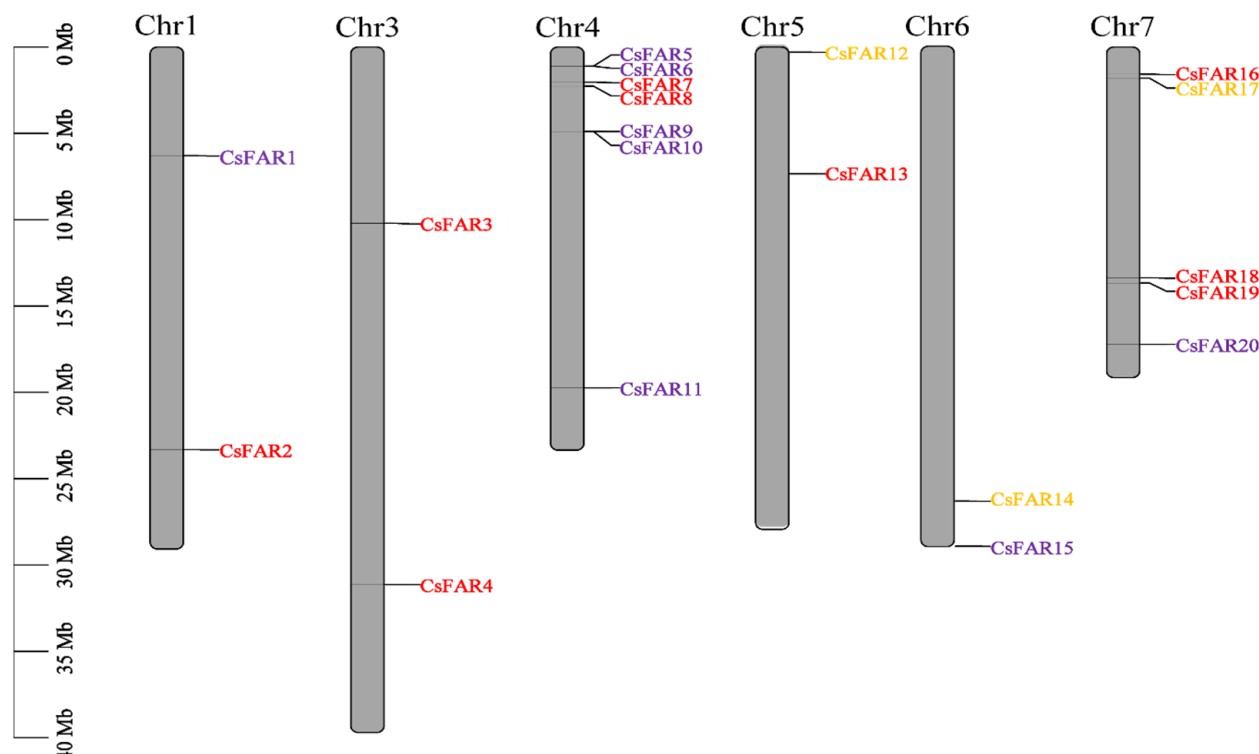

**Figure 3.** The distribution of *FAR1/FHY3* gene family members of chromosomes in cucumber. Each vertical line prominently displayed the number of chromosomes, while the gene names were conveniently presented on the right side of the corresponding chromosome. Yellow: *FAR1/FHY3* Class I; purple: *FAR1/FHY3* Class II; red: *FAR1/FHY3* Class III; chr: chromosome.

*3.5. Phylogenetic and Collinearity Analysis of FAR1/FHY3 Family Genes Cucumber*

According to the sequence similarity, 61 FAR1/FHY3 proteins can be divided into three groups (Figure 4). In Group A, there are three, two and five FAR1/FHY3 from cucumber, Arabidopsis and tomato, respectively. Group B included eight, seven and thirteen FAR1/FHY3 proteins from cucumber, Arabidopsis and tomato, respectively. A total of nine, five and nine FAR1/FHY3 members from cucumber, Arabidopsis and tomato were present in Group C. Among them, CsFAR12, CsFAR17 and CsFAR14 belong to group A and have the highest homology with AtFAR13, SlFAR19 and AtFAR12. CsFAR5, CsFAR20, CsFAR6, CsFAR15, CsFAR10, CsFAR9, CsFAR1 and CsFAR11 belong to Group B and have a closer relationship with AtFAR10, SlFAR9, SlFAR12, AtFAR6, AtFAR2, AtFAR1, AtFAR3 and AtFAR4. CsFAR3, CsFAR16, CsFAR2, CsFAR7, CsFAR8, CsFAR18, CsFAR4, CsFAR19 and CsFAR13 are in Group C and have upper homology with AtFAR14 and AtFAR9; AtFAR7, AtFAR5, SlFAR1 and SlFAR8; SlFAR14, SlFAR22, SlFAR23, SlFAR27 and CsFAR13; and SlFAR27 and CsFAR19 (Figure 4). The results of collinearity analysis showed that 20 cucumber genes are collinear with 14 Arabidopsis genes. There are seven homologous pairs between cucumber and Arabidopsis (Figure 5). Thus, cucumber *FAR1/FHY3s* are closely related to other plant *FAR1/FHY3* members.

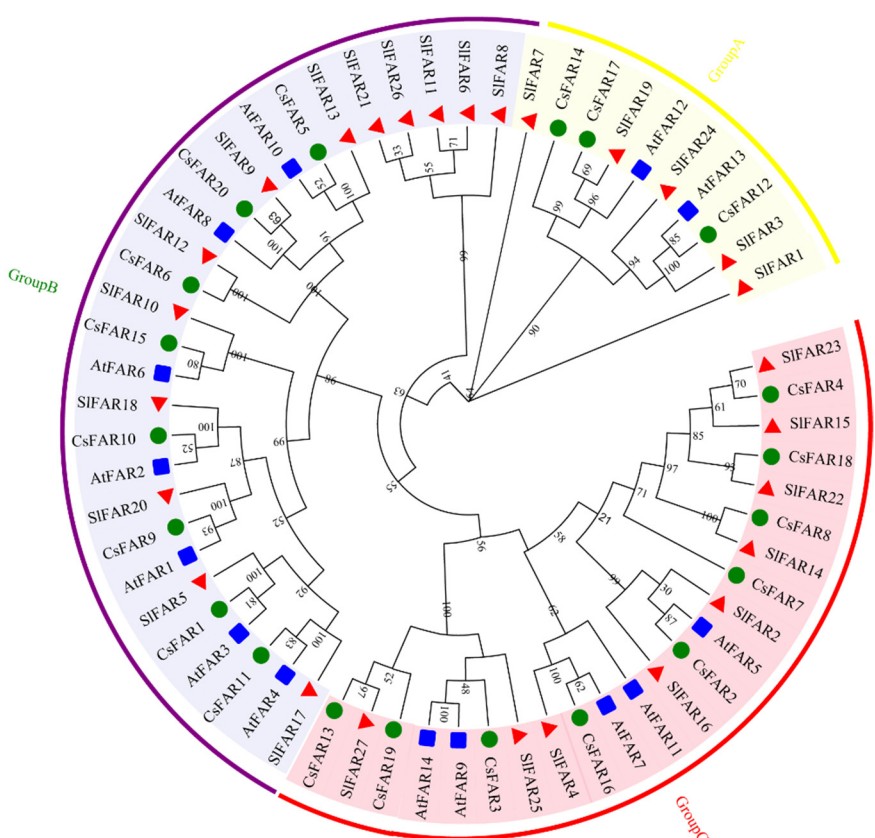

**Figure 4.** Evolutionary relationships of FAR1/FHY3 family in various species. To construct the phylogenetic tree, we employed the maximum likelihood method. This involved the inclusion of 20 cucumber (*Cucumis sativus* L.) FAR1/FHY3 proteins, 14 Arabidopsis (*Arabidopsis thaliana* L.) FAR1/FHY3 proteins, and 27 tomato (*Solanum lycopersicum* L.) FAR1/FHY3 proteins. The three subgroups are colored differently. The three differently colored shapes represent FAR1/FHY3 proteins from 3 species. The green circle, blue rectangle, and red triangle represent cucumber, Arabidopsis, and tomato FAR1/FHY3 proteins, respectively.

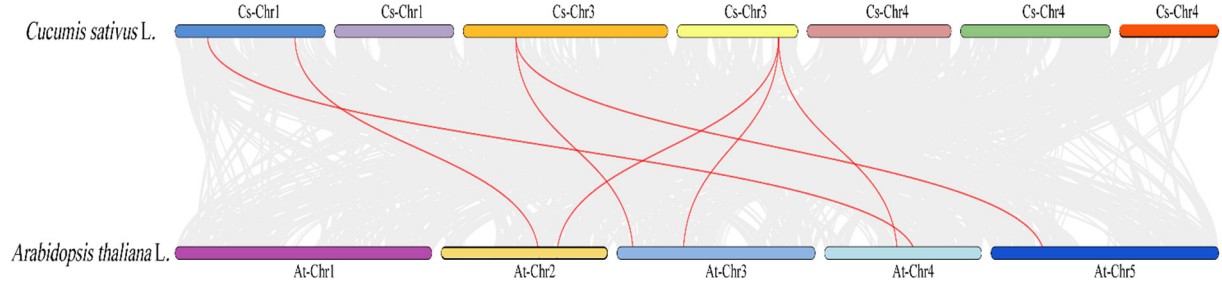

**Figure 5.** The collinearity analysis of the *FAR1/FHY3* gene family was conducted in *Cucumis sativus* L. and Arabidopsis thaliana L. Syntenic *FAR1/FHY3* gene pairs are represented by the marked red lines.

### 3.6. Cis-Element in FAR1/FHY3 Gene Promoter of Cucumber

The predicted cis-acting elements were classified as hormone signal response, abiotic stress response and light signal response according to the function and action of these elements (Table 5). The *CsFAR1/FHY3* gene contains 13 cis-elements that respond to hormones and stresses. There are several elements that respond to abiotic stress, namely the LTR, ARE, MBS and GC motif repeats. In terms of light signal response, we have circadian, G-box, $O^2$-site and TGA-element. Additionally, many of the *CsFAR1/FHY3* genes contain hormone signal responsive elements, such as AuxRR-core, CGTCA-motif, TATC-box, TCA-element and ABRE (Figure 6). As shown in Figure 6, ABRE elements are mostly

distributed in *CsFAR2*, *CsFAR4*, *CsFAR6*, *CsFAR8*, *CsFAR10*, *CsFAR11*, *CsFAR13*, *CsFAR17* and *CsFAR19*. CGTCA-motif elements are mostly distributed in *CsFAR3*, *CsFAR5*, *CsFAR10*, *CsFAR13*, *CsFAR16* and *CsFAR20*. TCA-element elements are mostly distributed in *CsFAR1*, *CsFAR9*, *CsFAR13*, *CsFAR14* and *CsFAR16*. GC-motif elements are mostly distributed in *CsFAR13*. LTR elements are mostly distributed in *CsFAR1*. MBS elements are mostly distributed in *CsFAR1* and *CsFAR6*. ARE elements are more distributed in *CsFAR2*, *CsFAR4*, *CsFAR7*, *CsFAR8*, *CsFAR10* and *CsFAR16*. G-box elements are more distributed in *CsFAR2*, *CsFAR4*, *CsFAR6*, *CsFAR8*, *CsFAR10*, *CsFAR11*, *CsFAR13*, *CsFAR17* and *CsFAR19*. $O^2$-site elements are mostly distributed in CsFAR16. Finally, TGA-element elements are mostly distributed in *CsFAR5*, *CsFAR8*, *CsFAR16* and *CsFAR18* (Figure 6). Through our examination of cis-regulatory regions, it came to light that a substantial portion of the *CsFAR1/FAY3* genes might wield significant influence over stress, light and hormone reactions.

**Table 5.** Summary of cis-acting elements of *CsFAR1/FHY3* genes.

| Element | Sequence | Description |
|---|---|---|
| ABRE | (C/T) ACGTG (G/T) | cis-acting element involved in the abscisic acid responsiveness |
| AuxRR-core | GGTCCAT | cis-acting regulatory element involved in auxin responsiveness |
| CGTCA-motif | CGTCA | cis-acting regulatory element involved in the Me-JA-responsiveness |
| TATC-box | TATCCCA | cis-acting element involved in gibberellin-responsiveness |
| TCA-element | CCATCTTTTT | cis-acting element involved in salicylic acid responsiveness |
| GC-motif | CCCCCG | enhancer-like element involved in anoxic specific inducibility |
| LTR | CCGAAA | cis-acting element involved in low-temperature responsiveness |
| MBS | CAACTG | MYB binding site involved in drought-inducibility |
| ARE | AAACCA | cis-acting regulatory element essential for the anaerobic induction |
| G-Box | CACGTG | cis-acting regulatory element involved in light responsiveness |
| $O^2$-site | GTTGACGTGA | cis-acting regulatory element involved in zein metabolism regulation |
| TGA-element | TGACGTAA | auxin-responsive element |
| circadian | CAAAGATATC | cis-acting regulatory element involved in circadian control |

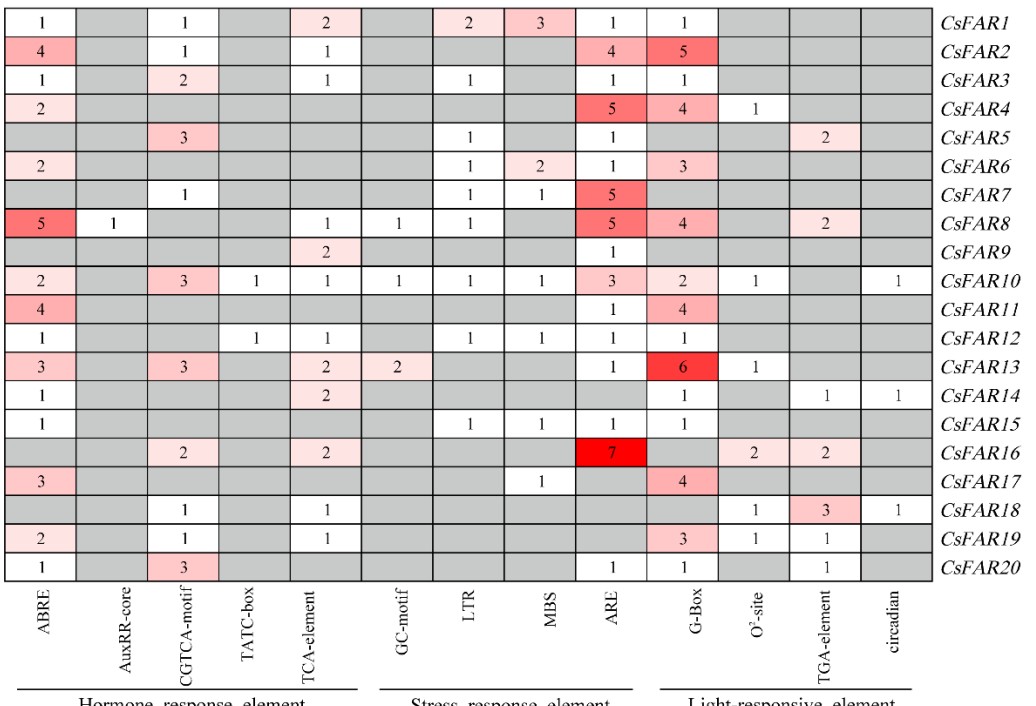

**Figure 6.** The quantities of elements in the *CsFAR1/FHY3* genes' upstream regions spanning 2000 bp. The grid's diverse colors and numbers represent the counts of distinct cis-acting regulatory elements within these *CsFAR1/FHY3* genes.

### 3.7. Expression Analysis of Cucumber FAR1/FHY3 Genes in Different Tissues

The expression results of *FAR1/FHY3* genes in different organs showed that the expression of all genes was highest in roots, which was significantly higher than that in stems and leaves (Figure 7). For example, compared with that in roots, the expression of *CsFAR10*, *CsFAR3*, *CsFAR1* and *CsFAR14* was decreased in stems by 54%, 56% 79% and 78%, respectively. The expression levels of *CsFAR5*, *CsFAR6*, *CsFAR14* and *CsFAR4* in leaves declined by 73%, 74%, 89% and 88%, in comparison with those in roots, respectively. Except *CsFAR5*, the expression of *FAR1/FHY3* genes was higher in stems than in leaves. For example, compared with that in leaves, the expression of *CsFAR3*, *CsFAR10*, *CsFAR4* and *CsFAR11* in stems was increased by 45.45%, 54.34%, 52% and 59.37%, respectively. Our results suggest that this gene is expressed at different levels in different tissues or organs, and it may play a specific role in the growth of cucumber seedlings.

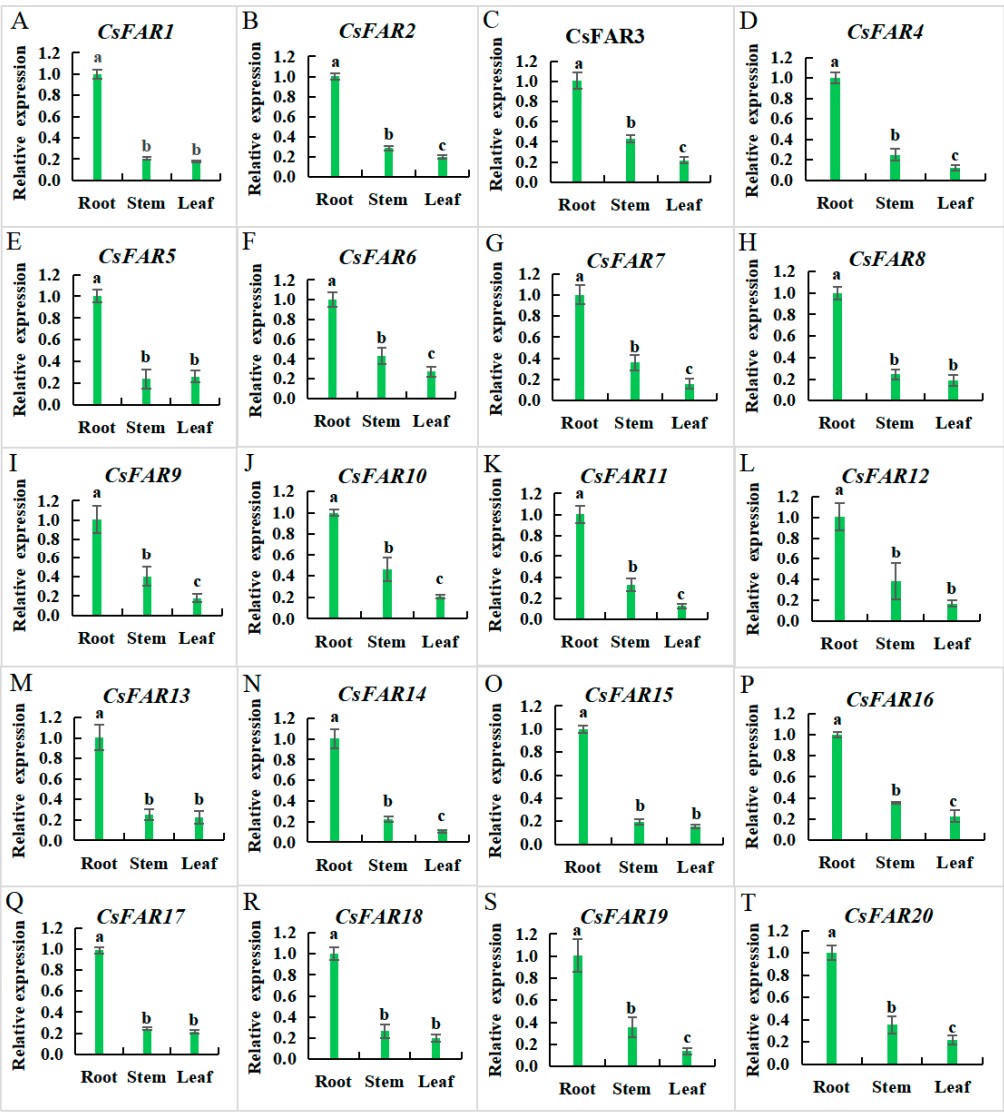

**Figure 7.** Expression levels of the *FAR1/FHY3* gene were analyzed in the roots, stems, and leaves of cucumber. The expression patterns of the *CsFAR1* and *CsFAR20* genes in various tissues are illustrated in figures (**A**–**T**) correspondingly. The error bars signify the standard error, which was calculated based on three independent replicates. The relative expression of each gene in distinct tissues is presented as the mean ± SE (n = 3). Bars labeled with different lowercase letters indicate significant differences, as determined by Duncan's multiple range tests ($p < 0.05$).

*3.8. Expression Analysis of Cucumber FAR1/FHY3 Genes under Different Treatments*

The relative expression of *CsFAR4*, *CsFAR9* and *CsFAR6* was significantly up-regulated under Me-JA treatment. After 6 h of Me-JA treatment, the expression of *CsFAR9* and *CsFAR14* reached the highest value and then gradually decreased. However, *CsFAR6* reached its highest expression level after 24 h of Me-JA treatment. The expression levels of *CsFAR12*, *CsFAR16*, *CsFAR13* and *CsFAR20* were significantly up-regulated after 6 h of ABA treatment and then decreased. The maximum expression value of *CsFAR19* was reached at 24 h. Interestingly, the expression level of *CsFAR20* under ABA treatment showed a gradual increase over time.

Under salt stress, the expression levels of *CsFAR8* and *CsFAR13* were significantly up-regulated at 6 h, where *CsFAR8* reached the highest value at 12 h then slowly decreased. *CsFAR13* reached the highest value at 48 h. The expression levels of *CsFAR12*, *CsFAR14*, *CsFAR10* and *CsFAR15* were significantly up-regulated under PEG treatment. Among them, the expression of *CsFAR12* was the highest at 6 h, and the expression levels of *CsFAR17* and *CsFAR19* remained unchanged in all periods. Under dark treatment, four genes (*CsFAR20*, *CsFAR5*, *CsFAR11* and *CsFAR8*) reached the highest expression levels at 12 h, and CsFAR15 and CsFAR13 reached the highest expression levels at 24 h. For *CsFAR2*, *CsFAR19* and *CsFAR17*, their expression levels exhibited a downward trend under all treatments.

## 4. Discussion

FAR1/FHY3 is a novel class of transcription factors derived from transposases that have been domesticated and adapted over time to form the FAR1/FHY3 transcription factor family [36]. A previous study suggested that FAR1/FHY3 mainly activates light-induced transcription of target genes [15]. A collection of 14 *FAR1/FHY3* genes with remarkable similarity in gene structure and crucial biological functions have been recognized in Arabidopsis [11]. These genes play an imperative role in sustaining the proper growth of plants. However, the *FAR1/FHY3* gene family in cucumber has not been studied in detail. In this study, we screened the members of the *FAR1/FHY3* family in the cucumber genome and found 20 in total (Table 2). The 20 genes of *FAR1/FHY3* gene family are distributed on six chromosomes of cucumber (Figure 3), indicating that they have their own different expression patterns. While 33 *FAR1/FHY3* members were identified on 11 chromosomes in *Eucalyptus grandis* [8], 246 *FAR1/FHY3* members were identified on 20 chromosomes in *Arachis hypogaea* [9]. It is evident that an increase in the quantity of chromosomes corresponds to a proportional increase in the number of *FAR1/FHY3* genes. The homology between the different genes implies that members of the *FAR1/FHY3* gene family underwent duplication or recombination events during the evolution of cucumber. Obviously, on chromosome 4, *CsFAR5*, *CsFAR6*, *CsFAR7*, *CsFAR8*, *CsFAR9* and *CsFAR10* closely pack together to form a gene cluster (Figure 3), suggesting that these five members may encode proteins and regulate the related biological processes together [37]. In addition, there were four tandemly duplicated gene clusters on chromosomes 4 and 7 (Figure 3), suggesting their functional similarity. The presence of tandem repeat genes has, to some extent, provided the raw material for the evolution of the FAR1/FHY3 transcription factor family [38].

Transcription factors primarily play a role in regulating gene expression within the nucleus [39]. However, some of the identified *CsFAR1/FHY3* family members were predicted to be located in other cell components, such as *CsFAR4*, *CsFAR6*, *CsFAR8*, *CsFAR10*, *CsFAR12*, *CsFAR17* and *CsFAR19* (Table 2). In Arabidopsis, the absence of a presumed nuclear localization sequence (NLS) has been predicted for *AtFRS1*, *AtFRS8* and *AtFRS9*. However, intriguingly, these factors can still effectively localize to the nucleus either through an unconventional NLS or by interacting with other protein counterparts [11]. This suggests that although some of the predicted *CsFAR1/FHY3* genes are not present in the nucleus, they may enter the nucleus by interacting with other members of *CsFAR1/FHY3* to form complexes. It was found that FHY3 and FAR1 in Arabidopsis bound to the FBS motif on the promoters of FHY1 and FHL, respectively, thereby activating the transcription of FHY1 and FHL [11]. Then, [40] found that FHY1 and FHL directly interacted with Pfr to assist

the entry of phy A into the nucleus. Therefore, FAR1/FHY3 may play a critical role in initiating the phy A signaling pathway, which directly determines whether phy A can enter the nucleus to perform biological functions after light induction.

In recent years, several studies have shown that *FAR1/FHY3* plays an important role in various physiological and developmental processes, such as plant structural function [41] and flowering [42], chlorophyll biosynthesis [43] and chloroplast generation [43], circadian clock carrying [44], UV-B signaling [45], ABA signaling [18], ROS homeostasis [46] and programmed cell death (PCD) [47]. In order to understand the relationship between *FAR1/FHY3* members of cucumber and other species, we constructed phylogenetic trees of *FAR1/FHY3* members of cucumber, Arabidopsis and tomato, and we divided 61 *FAR1/FHY3* members into three groups (Figure 4). Among them, the grouping of some genes changed considerably during the evolutionary process, such as *CsFAR14*, *CsFAR7*, *CsFAR19*, etc. It is hypothesized that these genes may have undergone loss and gain events during the evolutionary process, resulting in changes in their functions, and their DNA sequences disambiguated into sequences that can encode a new protein over a sufficiently long period of evolutionary time [48]. As shown, these three subgroups are highly similar to the Arabidopsis and tomato *FAR1/FHY3* families, suggesting that the FAR1/FHY3 transcription factor family is highly conserved during evolution [11]. It can be hypothesized that the cucumber *FAR1/FHY3* family and the Arabidopsis and tomato *FAR1/FHY3* family genes may have similar functions among the homologous genes. The functions of the corresponding genes in the cucumber *FAR1/FHY3* family can be predicted through the study of the functions of the Arabidopsis and tomato *FAR1/ FHY3* family genes [11].

Protein motifs and gene structures that remain conserved serve as vital foundations for genes' biological functioning. Within the scope of this research, it was observed that members of the *FAR1/FHY3* family demonstrate analogous protein motifs and gene structure characteristics (Figure 2). This finding is consistent with previous reports on Arabidopsis and tea plant [10,11]. We analyzed the gene structure of *FAR1/FHY3* and found that the number of introns in all *FAR1/FHY3* genes ranges from 1 to 8 (Figure 1). The prediction of the number of cis-acting elements indicated that the *CsFAR1/FHY3* gene family contains rich regulatory elements, such as G-box, TGA-element, $O^2$-site, AuxRR-core, CGTCA-motif, TATC-box, TCA-element and ABRE (Figure 6), suggesting a wide range of biological functions of the *FAR1/FHY3* family members in cucumber. The appearance of G-box in *CsFAR1/FHY3* gene family indicates that the *FAR1/FHY3* genes in cucumber might be involved in the photoreceptive response (Figure 6).

Previous studies have shown that the *FAR1/FHY3* genes exhibit different tissue-specific expression patterns in different plants [10,11]. Different expression patterns have been observed in rosette leaves, cauline leaves, stems, flowers and siliques for genes belonging to the *AtFAR1/FHY3* family. In contrast, *AtFRS10* was exclusively identified in hypocotyl and cotyledons [11]. The *AtFAR1* genes were expressed in leaves, stems and flowers [11]. In tea plants, most *FAR1/FHY3* genes were highly expressed in leaves and shoots but not in stems and roots [10]. In this study, tissue-specific expression analysis of the *FAR1/FHY3* gene family showed that almost all members have high expression levels in roots, followed by stems and leaves (Figure 7). Thus, *FAR1/FHY3* gene family numbers have different expression patterns in different tissues, indicating that they might exert different regulatory roles in plant growth, development and stress response.

A significant expression of *FAR1/FHY3* in response to biotic/abiotic stresses was found in Arabidopsis, indicating that the *AtFAR1/FHY3* gene might play an important role in plant response to adverse conditions [45]. A previous study showed that more than half of downstream gene families, such as $C_3H$, WRKY and FAR1, were up-regulated under salt stress, revealing that these transcription factor families may participate in salt response [49]. The qRT-PCR results showed that most members could be induced by NaCl and PEG6000 stresses and inhibited by ABA (Figure 8). In particular, *CsFAR14*, *CsFAR10* and *CsFAR18* genes were up-regulated under PEG6000 stress, whereas they were significantly suppressed under ABA treatment. This phenomenon may be attributed to the existence of a feedback

inhibitory mechanism that regulates gene expression in plants. Because abiotic stresses such as salt stress and osmotic stress significantly induced the expression of *FAR1/FHY3* genes, the continuous induction of ABA in the plant would inhibit the sustained up-regulation of these genes to a certain extent, so that a dynamic equilibrium could be achieved to withstand the external stress conditions [18]. Therefore, cucumber's *FAR1/FHY3* genes exhibit diverse expression in response to both hormonal and abiotic stresses, suggesting their potential involvement in distinct cellular processes and cell signaling mechanisms.

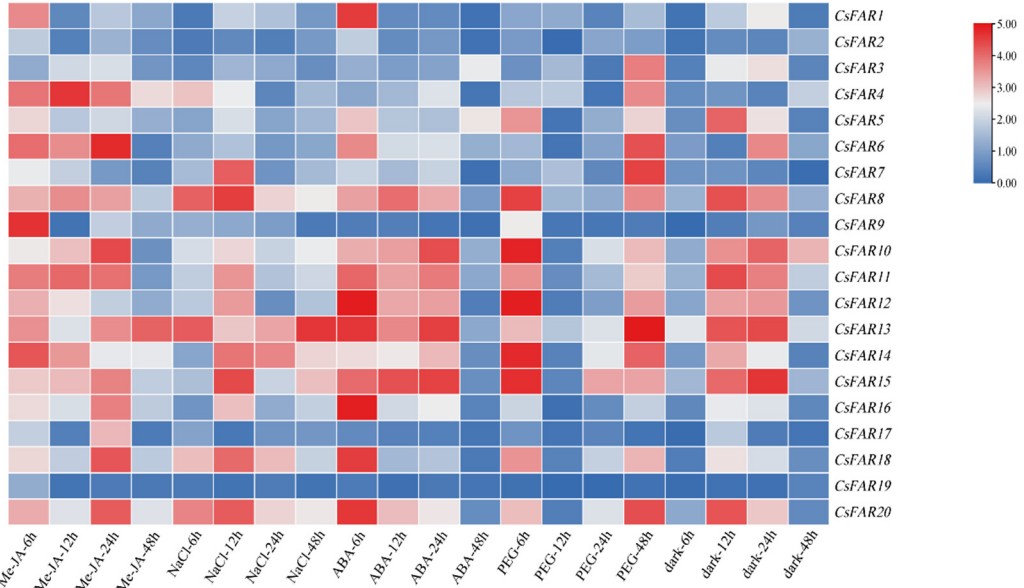

**Figure 8.** Expression leaves of *FAR1/FHY3* genes in cucumber under ABA, Me-JA, NaCl, dark and PEG6000 treatments. Seedlings were treated with 100 μM ABA, 100 μM Me-JA, 50 mM NaCl, dark and 8% (*w/v*) PEG. The color scale represents the fold change normalized by log2 converted data. Blue shows down-regulated genes and red shows up-regulated genes.

## 5. Conclusions

In this study, the *FAR1/FHY3* gene family in cucumber was comprehensively and systematically analyzed. A total of 20 *FAR1/FHY3* cucumber genes were identified and analyzed for their gene structures, conserved motifs, phylogenetic relationships, gene locations and cis-elements. Different plants have conserved the **FAR1/FAY3** genes. The involvement of *CsFAR1FHY3* in regulating photomorphogenesis, growth, development and abiotic stresses is suggested through its modulation of downstream responses. Overall, these analyses will facilitate the functional study of *FAR1/FHY3* genes in cucumber biology and provide strong support for further exploration of the involvement of *FAR1/FHY3* genes in plant growth and development and adversity response.

**Author Contributions:** Formal analysis, X.L. and W.L.; funding acquisition, C.W. and W.L.; software, Y.L., S.L. and K.Y.; visualization, Y.Q. and W.L.; writing—original draft, X.L.; writing—review and editing, X.L. and W.L. All authors have read and agreed to the published version of the manuscript.

**Funding:** This work was supported by the National Natural Science Foundation of China (Nos. 32360743, 32072559, 31860568, 31560563 and 31160398); the Key Research and Development Program of Gansu Province, China (No. 21YF5WA096); the National Key Research and Development Program (2018YFD1000800); the Research Fund of Higher Education of Gansu, China (No. 2018C-14 and 2019B-082); and the Natural Science Foundation of Gansu Province, China (No. 1606RJZA073). The funders had no role in study design, data collection and analysis, decision to publish or preparation of the manuscript.

**Data Availability Statement:** All relevant data are contained within the article.

**Conflicts of Interest:** The authors declare no conflict of interest.

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
