# Peer review of "Genome-Wide Identification and Expression Analysis of FAR1/FHY3 Gene Family in Cucumber (Cucumis sativus L.)"

_agronomy, doi:10.3390/agronomy14010050_

Round 1

Reviewer 1 Report

Comments and Suggestions for Authors

82: Check it....First of all, from the Ensembl the Plants the Cucumber genome database c

139: Check it....Cucumber (C. sativus L. ‘Xin Chun 4’) 

Author Response

Dear Reviewer,

Thanks a lot for having reviewed our manuscript (agronomy-2680632). We have revised the manuscript, and would like to submit it for your consideration. According to your comments and suggestions, we have made corresponding changes. The revisions have been highlighted in the revised manuscript.

I greatly appreciate both your help and that of the referees concerning improvement to this paper. We hope that the revised version of the manuscript is now acceptable for publication in your journal.

I look forward to hearing from you soon.

We would like to express our sincere thanks again to you for the constructive and positive comments.

With best wishes,

Yours sincerely,

Xuelian Li, Weibiao Liao

Reviewer 2 Report

Comments and Suggestions for Authors

The authors did a great effort to obtain all these results. 

In the next few lines I would like to express my comments/questions:

Line 36: Correct to  "IMPAIRED RESPONSE".

Line 37: Another mistake: "FAR-RED ELONGATED HYPOCOTYL1 (FHY3)" 

Line 38: Why Arabidopsis is not written in italics? 

Line 39: "phytochrome A" I think it should be abbreviated as phyA. Please check.

Line 76: "different issues" should be corrected to "different tissues".

Line 82: Name of the database is Ensembl Plants.

Line 83: Correct to "were downloaded".

Line 84 - where you start with the description how gene members were identified: This section should be better described. How many FAR1/FHY3 gene members were already known in the ASM407v2 genome? All other orthologous sequences, I believe, were identified based on A. thaliana homologs? A. thaliana has 14 FAR1/FHY3 gene members. Did you take into account also the query coverage for filtering BLAST results (or any other BLAST parameters)?

Also in line 93 you mentioned bidirectional BLAST. How exactly did you perform bidirectional BLAST? Did you check 20 gene members with blasting against NCBI nr database? Is BLAST GUI Wrapper a wrapper for locally installed BLAST? I see that xml file was obtained with BLAST and then results were converted to table. Did you take only the top first hit for each query to make table?

Line 116: Please explain "Ensebml was prepared...".

Line 135: Please explain: "The relative FASTA...".

Line 139: C. sativus should be italics.

Line 161: Which parameters did you use for primer design?

Line 175: "The CsFAR genes are distributed..."?

Line 213: Conserved motifs are listed in Table 4 and not in Table 3.

Line 218: Why motifs in Table 4 are not sorted according to the order they appear in the proteins? 

Figure 6: Correct to "Hormone response element".

Few additional questions: I'm interested why tools, which take into account the whole proteoms (like OrthoFinder) was not employed for this study? Is the number of species included (in addition to Cucumis sativus) adequate?

Author Response

(The authors gave the same response as above.)

Reviewer 3 Report

Comments and Suggestions for Authors

I reviewed the manuscript “Genome wide Identification and Expression Analysis of FAR1/FHY3 Gene Family in Cucumber (Cucumis sativus L.)” submitted to the journal Agronomy.

My comments are:

Line 66: Please, add abscisic acid (ABA) instead of ABA because this is the first time to mention it.

Line 70: please, add a short paragraph regarding the economic importance of cucumber including the global productivity, global harvested area, nutrients….etc.

Line 77: “abiotic acid” do you mean abscisic acid?

Line 139: Why did you soak the cucumber seeds at 55 â—¦C for about 139 4 h?

Line 142: What are the components of Yamazaki cucumber nutrient solution?

Line 146: Why did you used these concentrations specifically? Why did not you use more than one concentration of each treatment?

Line 213: Table 4 NOT 3

Line 230: Figure 2B, Amino acid sequences of different conserved motifs, is not clear. Kindly, modify it.

Line 252: Why did you use tomato also, I think Arabidopsis is enough.

Comments on the Quality of English Language

Minor editing of English language required

Author Response

(The authors gave the same response as above.)

Reviewer 4 Report

Comments and Suggestions for Authors

The study in the submitted article aimed to identify and to investigate FAR1/FHY3 genes in cucumber. One of the objectives of the study was the classification of identified 20 genes. Also, authors investigated the chromosomal location of this genes, physical and chemical properties, location, structure.
The introduction explains very well why this experiment was necessary to be carried out. The hypotheses are well established. Everything is clearly described. The material and methods are described in great detail. Everything is clear and sufficient information is provided about the setting of the experiment.

The results are clearly displayed. The results have been well analyzed. In the discussion, the authors link to each segment of their experiment and supplement their findings with arguments. As the authors stated, their results are first in this field, an experiment was carried out on an important plant species and obtained results are great introduction to a future research.

The cited references are relevant to the experiment. Out of 47 cited references, 28 were published within 10 last years (average within 8.9 years). Only 4 of 47 (8, 25, 29, 35) are auto citations, and they are relevant to the subject.

Specific comments:

Line 70: Cucumber is not melon crop. They belong to the same genus. Please correct this sentence.

Line 175: “CsFAR gene” – please replace with CsFAR genes

Line 298: please replace "rewponse" with response

Author Response

(The authors gave the same response as above.)
